# Impact of Gut Microbiome Modulation on Uremic Toxin Reduction in Chronic Kidney Disease: A Systematic Review and Network Meta-Analysis

**DOI:** 10.3390/nu17071247

**Published:** 2025-04-03

**Authors:** Renata Cedillo-Flores, Miguel Angel Cuevas-Budhart, Iván Cavero-Redondo, Maria Kappes, Marcela Ávila-Díaz, Ramón Paniagua

**Affiliations:** 1Unidad de Investigación Médica en Enfermedades Nefrológicas, Centro Médico Nacional Siglo XXI, Ciudad de México 06720, Mexico; renatta27d16@gmail.com (R.C.-F.); cramav@gmail.com (M.Á.-D.); 2CarVasCare Research Group, Facultad de Enfermería de Cuenca, Universidad de Castilla la Mancha, 16002 Cuenca, Spain; ivan.cavero@uclm.es; 3Faculty of Healthcare Sciencies, Nursing School, Universidad San Sebastián, Puerto Montt 5501842, Chile; maria.kappes@uss.cl

**Keywords:** chronic kidney disease, probiotics, prebiotics, synbiotics, gut microbiota, uremic toxins

## Abstract

**Background/Objectives**: Chronic kidney disease is associated with increased intestinal barrier permeability, leading to heightened inflammation and oxidative stress. These changes contribute to complications such as cardiovascular disease, anemia, altered mineral metabolism, and CKD progression. Interventions using prebiotics, probiotics, and synbiotics may mitigate dysbiosis and improve intestinal barrier function, Under this premise, the objective of this network meta-analysis was to evaluate the effect of probiotics, prebiotics, and synbiotics in reducing uremic toxins produced by the gut microbiota in CKD patients. **Methods**: A systematic review and network meta-analysis of randomized clinical trials (RCTs) was performed in the following databases: Web of Science, Scopus, the Cochrane Register of Controlled Trials, and PubMed published between 2019 and 2023. The analysis focused on the use of prebiotics, probiotics, and synbiotics in CKD patients at stages 3 to 5, as per KDIGO guidelines, and their association with reductions in uremic toxins such as Indoxyl Sulfate, p-Cresyl Sulfate, urea, and creatinine. The risk of bias was assessed using the Cochrane risk of bias tool (RoB 2), with evaluations conducted independently by two reviewers, and a third consulted for disagreements. The study follows the PRISMA statement. **Results**: The studies included 331 patients, primarily male, across CKD stages 3a to 5. The interventions positively impacted the gut microbiota composition, leading to reductions in free and total p-Cresyl Sulfate (SUCRA: 72.6% and 66.2, respectively) and indoxyl sulfate (SUCRA: 88.5% and 83.1%). **Conclusions**: The findings suggest that modulating the gut microbiota through these interventions can effectively reduce specific uremic toxins. However, further trials are necessary to better understand microbiota modulation and its impact on intestinal bacterial composition (PROSPERO number: CRD42023438901).

## 1. Introduction

Chronic kidney disease (CKD) is a progressive condition characterized by the gradual loss of kidney function over time. It affects approximately 10% of the global population, with higher prevalence rates in older adults and individuals with comorbidities such as diabetes and hypertension [1]. CKD is associated with increased intestinal barrier permeability, leading to heightened inflammation and oxidative stress, which contributes to complications such as cardiovascular disease, anemia, and altered mineral metabolism [2].

Recent research has highlighted the role of gut microbiota in the development and progression of CKD. Dysbiosis, or the imbalance of gut microbiota, is common in CKD patients and is linked to the accumulation of uremic toxins such as Indoxyl Sulfate (IS) and p-Cresyl Sulfate (pCS) [3]. These toxins exacerbate CKD and its complications by promoting inflammation and vascular dysfunction. Interventions using prebiotics, probiotics, and synbiotics have been explored to modulate gut microbiota, aiming to reduce uremic toxin levels and improve patient outcomes [4,5].

Various conditions present in patients with chronic kidney disease (CKD) are associated with the development of alterations in the permeability of the intestinal barrier and the gut microbiome (GM), among which are the loss of renal function, uremic toxicity, and the frequent use of antibiotics [6].

Changes in the quantitative and qualitative composition of the intestinal microbial population have been implicated in the pathogenesis of different diseases, including the systemic inflammatory state, CKD progression, and CKD-related cardiovascular complications. This is highly dependent on substantial changes, for example, in diet composition and nutrient intake due to recommended restrictions to prevent the complications [7,8].

It is important to add that the GM changes constantly during CKD, producing a metabolic load that could further increase cardiovascular risk. In addition, metabolites derived from the gut microbiota, including products of protein or choline fermentation, such as p-Cresyl sulfate (pCS) and indoxyl sulfate (IS), may contribute to decreased renal function and worsen kidney function and cardiovascular disease [7].

### 1.1. Probiotics, Prebiotics, and Synbiotics

Probiotics are live microorganisms that, when administered in adequate amounts, confer a health benefit on the host [9,10]. On the other hand, prebiotics are mostly fibers that are indigestible food ingredients and beneficially affect host health by selectively stimulating the growth or activity of some genera of microorganisms in the colon, typically lactobacilli and Bifidobacteria. Lastly, synbiotics are next-generation probiotics made with various formulations of probiotics and prebiotics that work synergistically to restore healthy gut ecology [11,12,13].

In this sense, immunonutrition can be defined as the area of nutrition dedicated to studying the processes by which nutrients modulate the actions of the immune system and their use for this purpose. This encompasses nutrition, infection, inflammation, tissue damage, and immunity, with a key role in the immune, endocrine, nervous, and microbiota systems [14,15].

### 1.2. Uremic Toxins, CKD, and the Risk of Cardiovascular Disease

P-Cresyl sulfate can be considered a prototypical protein-bound uremic toxin, and IS a circulating uremic toxin. Both are excreted via the kidneys, and serum concentrations increase progressively as GFR decreases. These metabolites cause vascular endothelial cell injury by increasing leukocyte activation and adhesion and contributing to local inflammation and oxidative stress [16,17].

Furthermore, higher concentrations of circulating IS and pCS correlate with measures of vascular dysfunction and aortic calcification and are independently associated with cardiovascular disease among individuals with kidney disease [16,17].

GM is altered in CKD patients due to increased intestinal permeability and the accumulation of uremic toxins in plasma (causing vascular alterations). Various therapeutic interventions, such as prebiotics, probiotics, and synbiotics, have recently been explored to improve intestinal microbiota dysbiosis. These could reduce the generation of uremic toxins by increasing or decreasing the associated bacteria, reducing the cardiovascular effects that CKD brings with it [2].

Some related studies in this context are Pan et al. [18], who found a decrease in C-reactive protein (CRP) and IL-6 levels after two months of treatment in 58 patients treated with probiotics; in addition, higher levels were obtained in the domains of physical functioning and social functioning than in the control group.

In another study, Raquel Armani et al. [19] found that the prebiotic FOS reduced circulating levels of IL-6 in patients with CKD and preserved endothelial function only in those with less damaged endothelium. No effect of FOS on arterial stiffness was observed.

This work aims to analyze the relationship between probiotics, prebiotics, and synbiotics in reducing uremic toxins produced by the intestinal microbiota.

## 2. Materials and Methods

### 2.1. Design

A systematic review and network meta-analysis of randomized clinical trials (RCTs) was conducted. A systematic and rigorous process was followed for the identification and evaluation of the existing scientific evidence on the relationship between the use of prebiotics, prebiotics, and synbiotics in patients with chronic kidney disease in stages 3 to 5 according to the KDIGO guidelines and the reduction of uremic toxins: Indoxyl Sulfate (IS), p-Cresyl Sulfate (pCS), Urea, Creatinine (Cr), and Phosphate (PHOS).

The recommendations of the PRISMA statement and the Cochrane Manual for systematic reviews and meta-analyses of intervention studies were followed [20,21].

First, the PICO strategy was used to develop the research question: What is the effectiveness of using probiotics, prebiotics, and synbiotics to reduce uremic toxins produced by the intestinal microbiota in patients with CKD stages 3 to 5?

### 2.2. Eligibility Criteria

The inclusion criteria for the review were randomized controlled trials published between 2019 and 2023, available in open access, and published in English or Spanish. The studies needed to be related to the use of prebiotics, probiotics, and synbiotics in patients with chronic kidney disease (CKD) in stages 3 to 5, focusing on the reduction of uremic toxins such as IS, pCS, Urea, Creatinine, and PHOS. The exclusion criteria involved studies that were unrelated to the topic, did not use prebiotics or probiotics, or did not involve CKD patients.

Studies were grouped for synthesis based on the type of intervention: prebiotics, probiotics, and synbiotics. The interventions were further analyzed for their effects on reducing specific uremic toxins.

### 2.3. Information Sources

The databases searched to identify relevant studies included Web of Science, Scopus, the Cochrane Register of Controlled Trials, and PubMed. These databases were chosen for their comprehensive coverage of scientific literature, particularly in the fields of health and medicine. The search strategy utilized Health Sciences Descriptors (DeCS) and Medical Subject Headings (MeSH) to ensure the inclusion of relevant studies. Keywords used in the search included gut microbiota, gut microbiome, chronic kidney disease, cardiovascular disease, and clinical trial. The document does not specify the exact dates when each database was last searched, which is a limitation regarding replicability and transparency.

### 2.4. Search Strategy

An extensive search was conducted across multiple databases, focusing on clinical trials involving individuals with CKD who received interventions such as prebiotics, probiotics, or oral synbiotics. The objective was to identify studies examining the impact of these interventions on the gut microbiota and their potential to reduce uremic toxins, which are known to accumulate in CKD patients and contribute to disease progression.

The search strategy utilized Boolean operators to refine the results. The operator “AND” established logical connections between different concepts, ensuring that retrieved documents met all specified criteria. Conversely, the operator “OR” broadened the search, allowing for the inclusion of documents containing at least one of the specified keywords. This approach facilitated a comprehensive retrieval of relevant literature.

No specific filters or limits were applied beyond the inclusion criteria, which focused on randomized controlled trials (RCTs) published between 2019 and 2023. The studies had to be available in open access and published in English or Spanish. Studies not related to the topic were excluded from the review.

The strategy adhered to the PRISMA guidelines and the Cochrane Manual for systematic reviews as a part of a systematic review and network meta-analysis. This rigorous approach ensured a thorough evaluation of the existing scientific evidence on the relationship between gut microbiota-modulating interventions and the reduction of uremic toxins in CKD patients.

### 2.5. Selection Process

The study selection process was systematic and rigorous. Two independent reviewers initially screened records to ensure objectivity and reduce bias, with a third reviewer resolving any discrepancies. Automation tools were employed to mark ineligible records, streamlining the initial screening process by reducing the manual workload.

For data collection, two reviewers independently gathered data from each report to ensure accuracy and reliability, minimizing bias and errors. When clarification or additional information was needed, reviewers contacted study investigators.

Outcomes and Variables: Data were gathered for outcomes related to lowering uremic toxins, particularly IS, pCS, urea, creatinine (Cr), and phosphate (PHOS). Results for each outcome domain from every study were compiled, including various measures, time points, and analyses. In cases where multiple results were available, the most thorough and relevant data were highlighted.

Additional variables for which data were collected included participant characteristics (e.g., age, gender, CKD stage), intervention specifics (e.g., type, dose, duration of prebiotics, probiotics, or synbiotics), and funding sources. Assumptions regarding missing or unclear information were made based on the context provided in the studies, and attempts were made to contact study authors for clarification when necessary.

### 2.6. Assessment of Methodological Quality and Risk of Bias

The risk of bias in the included studies was assessed using version 2 of the Cochrane risk of bias tool for randomized trials (RoB 2) [22]. This tool is organized around a fixed set of bias dimensions, emphasizing different aspects of study design, conduct, and reporting. Each study was evaluated by at least two reviewers working independently to ensure objectivity and minimize bias. In instances of disagreement, a third reviewer was consulted to reach a consensus. The RoB 2 tool assesses several domains, including selection bias (randomization sequence and allocation concealment), performance bias (blinding of participants and personnel), detection bias (blinding of outcome assessors), attrition bias (incomplete outcome data), and reporting bias (selective reporting of results) [23].

### 2.7. Statistical Analyses

For each outcome, the effect measures used in the synthesis or presentation of results included mean differences for continuous outcomes such as the levels of uremic toxins (IS, pCS, Urea, Creatinine, and Phosphate). These measures were employed to compare the effectiveness of prebiotics, probiotics, and synbiotics in reducing uremic toxins in patients with chronic kidney disease. The results were pooled using a random-effects model to accommodate variability among studies, and the pooled estimates were presented with means and standard deviations. Additionally, the surface under the cumulative ranking (SUCRA) was calculated for each intervention to provide a numerical ranking of their effectiveness, with values ranging from 0 to 100, where higher values indicate better performance.

To assess the risk of bias due to missing results in a synthesis, which may arise from reporting biases, comparison-adjusted funnel plots were utilized. These plots help identify asymmetries that could suggest the presence of publication bias or other reporting biases. By plotting the effect sizes against a measure of study precision, such as the standard error, these plots can visually indicate whether smaller studies with non-significant results are missing from the analysis. Additionally, the network meta-analysis was conducted using a random-effects model, which helps account for variability among studies and can mitigate the impact of reporting biases on the overall synthesis.

The methodological quality of the included studies was assessed after their selection according to the JADAD criteria [24]. This scale evaluates the CS from 0 to 5, depending on the randomization and its method, the type of blinding and its method, and the losses and withdrawals from the study, considering low methodological quality for those with a score of less than 3.

### 2.8. Synthesis Methods

Eligibility for Synthesis: The process of determining which studies were eligible for synthesis involved tabulating the characteristics of study interventions and comparing them against the planned groups for each synthesis. This included categorizing studies based on the type of intervention (prebiotics, probiotics, or synbiotics) and the specific outcomes measured, such as the reduction of uremic toxins.

The synthesis of results was conducted using a network meta-analysis, which allowed for the comparison of multiple interventions simultaneously [25,26]. A random-effects model was employed to account for variability among studies. This decision assumed that true effects may differ across studies due to variations in study populations and methodologies. The software package used for this analysis was STATA version 16.0 (StataCorp, College Station, TX, USA).

To explore potential causes of heterogeneity among study results, subgroup analyses and meta-regression were performed. These methods helped identify factors that could contribute to variations in effect sizes, such as differences in study design or participant characteristics.

Sensitivity analyses were conducted to evaluate the robustness of the synthesized results. This involved calculating standardized residuals and excluding outliers with *p*-values greater than 0.05 based on the t-distribution. These analyses ensured that the findings were not disproportionately influenced by any study or outlier.

As study designs and outcome definitions varied, the random effect was used to pool estimates from studies. Pooled estimates were transformed and presented with means and standard deviations. As a sensitivity analysis, standardized residuals were calculated, and outliers with *p* > 0.05 (based on the t-distribution) were removed. Meta-regression and odds ratios (OR) with 95% confidence intervals (CI) were used.

Additionally, we estimated the surface under the cumulative ranking (SUCRA) for each intervention. SUCRA involves the assignment of a numerical value between 0 and 100 to simplify the classification of each intervention in the rankogram. The best intervention obtained a SUCRA value closest to 100, and the worst intervention obtained a value closest to 0 [27].

## 3. Results

### 3.1. Data Availability and Study Characteristics

After conducting thorough screenings and an in-depth evaluation of the articles that met the inclusion criteria, 13 documents were initially selected for assessment of methodological quality and risk of bias using the ROB2 tool. However, only 7 studies were ultimately included in the Network meta-analysis due to further eligibility criteria and quality considerations (Figure 1).

Table 1 shows that the selected studies found Italy [28,29,30] as the principal country with more clinical trials in CKD patients, followed by Brazil [28,31]. The interventions carried out were three with prebiotics [19,31,32], two with probiotics [15], and two with synbiotics [19,31,32,33,34].

The included studies were primarily single-center [19,28,29,31,32,33] and only one with two tertiary renal care outpatient departments. The total sample was 331 patients. Regarding the gender of the participants, there was a higher prevalence of men compared to women.

All included only adults with the KDOQI classification who did not participate in dialysis treatment. The patients enrolled in the studies were in stages 3a to 5, of which only one was performed with stage 3a patients [19] and one from 4 to stage 5.

Regarding the interventions, Ramos et al. [31] and Armani et al. [19] used prebiotics in sachet presentation with 12 g each, and Ebrahim et al. [32] used a dose of 13.5 g. The intake was after food (lunch or dinner), since they dissolved in water. Armani et al. [19] used a prebiotic two times a day (6 g each) and recommended using it added to food (smoothies, cereal, yogurt, etc.). It should be emphasized that Ramos et al. [31] progressively raised the dose of prebiotics to avoid gastrointestinal symptoms, starting with 3 g and increasing by 3 g every three days until reaching 12 g.

McFarlane et al. [34] and Cosola et al. [30] used synbiotics. The first consisted of 20 g/day of a high-resistant starch fiber supplement and the probiotic component taken in the morning with food, and the second of two bags/day of a mixture of probiotics, prebiotics, and natural antioxidants. McFarlane also had a two-week dose escalation in which the daily dose was 10 g and then 20 g.

Lastly, the use of probiotics was carried out by De Mauri et al. [29] and Simeoni et al. [28]; the first used strains of Bifidobacterium and Lactobacillus, administered in two doses per day for one month and one dose for the following two months. In the case of Simeoni, the treated group underwent three phases of treatment using different probiotics: intestinal cleaning, intestinal colonization, and microbiota maintenance. Phase 1 used one capsule during main meals for one week of a complex of probiotics. The intestinal colonization lasted for two weeks with one capsule of a complex of Bifidobacteria and one capsule of Lactobacillus once a day, and phase 3 used both Bisifelle and Ramnoselle oral administration, one capsule of each, twice per day during breakfast and dinner for three months.

The studies’ follow-up periods to evaluate the effect of the interventions were two months for the shortest [30], followed by two and a half months [32], three months [19,31], one year [34], and the longest of three years [29]. During the follow-up, the patients were evaluated at the beginning of the study, during the treatment, and at its end, except with Simeoni, who only evaluated patients at the beginning and end of the study.

Regarding biochemical parameters that had changes exclusively in the intervention group, HDL-C (mg/dL) decreased in two studies (*p* < 0.01) [19,31], as did LDL-C (*p* < 0.05) [32]. Also, azotemia, the CaxP [33], mean urinary indican, β2-microglobulin, mean C-reactive protein (CPR) [19], ionized calcium, albumin, alkaline phosphatase, and IL-6 [19] decreased significantly at the end of the treatments, all with a *p* < 0.04. Otherwise, serum sodium [19], serum iron level, and mean transferrin saturation (TSAT) [19] increased, all present at *p* < 0.005. The relative abundance of Bifidobacterium animalis and unclassified Blautia [34] and mean fecal Lactobacillales and Bifidobacteria concentrations also increased [19].

### 3.2. Risk of Bias in Studies

Of the seven studies that underwent a risk of bias assessment, five scored ‘some concerns’ and two ‘low’ [19,31,32]. The biases that were assessed were selection (randomization sequence and concealment of allocation sequence), performance (masking of participants and personnel), detection (blinding of outcome assessors), attrition (incomplete outcome data), and notification (selective notification of results).

All the studies analyzed were interventions with the intention of treating patients. The seven studies analyzed showed excellent randomization of the patients in the control and intervention groups. The software selected the patients and assigned them to each group. The item with the highest risk of bias was information on data missing from the trials.

### 3.3. Network Meta-Analysis (Immunonutrition and Uremic Toxins)

The network diagram for each of the uremic toxins in CKD stage 3 and 5 patients for each intervention group is shown in Figure 2.

Table 2 summarizes the comparisons of the SUCRA values and provides a relative classification of immunonutrition (probiotics, prebiotics, and synbiotics) for each of the uremic toxins. The plots of the area under the curve for each uremic toxin are found in Figure 3.

Prebiotics and probiotics were identified as the most effective options for reducing uremic toxins. Specifically, prebiotics showed a SUCRA value of 72.6% for pCS, while probiotics had a SUCRA value of 66.2%. For free p-CS, the SUCRA values were 78.9% for prebiotics and 63.8% for probiotics. Probiotics were particularly effective for both IS and free IS, with SUCRA values of 88.5% and 83.1%, respectively. Additionally, prebiotics demonstrated a SUCRA value of 74.6% for reducing urea levels.

There was no indication of potential effects on reducing uremic toxins such as phosphate and creatinine or on improving the glomerular filtration rate (Figure 4).

### 3.4. Individual Immunonutrition and Reduction of Uremic Toxins

The implementation of prebiotics in the diet with interventions was based on patients without renal function replacement therapy and stages 3 and 5 according to international guidelines. Said follow-up was carried out over 12 months. A reduction in the accumulation of uremic toxins was found in these patients compared to placebos.

The treatment effect size in the pCS population was higher for probiotics (−8.95 95% CI: −53.3, 35.0), although this reduction was insignificant. Similarly, when comparing total IS, a more significant effect size was observed for the probiotic group. However, the reduction was insignificant (−6.39 95% CI: −16.0, 3.75) (Figure 3).

The size of the nodes is proportional to the sample size of each immunonutrition intervention. Each uremic toxin has a combination of interventions. For example, for the uremic toxin pCS Total, prebiotics (*n* = 70), probiotics (*n* = 24), synbiotics (*n* = 28), as well as the thickness of the lines is inversely proportional to the number of studies available. The number of studies for each immunonutrition intervention was as follows: Total pCS (*n* = 4), free pCS (*n* = 4), IS (*n* = 5), free IS (*n* = 5), and Urea (*n* = 5). About the type of intervention, three articles with the use of prebiotics, two with the use of probiotics, and two with the use of synbiotics are shown.

In relation to the use of probiotics, there was a decrease in uremia toxins with SUCRA values for total IS (88.5%), free IS (83%), including intervention for the use of prebiotics for p-CS (72.4%), free IS (78.9%), and finally, urea (74.6%).

Appendix A provides an individual presentation of each study, organized by treatment, and features a network meta-analysis. This material offers data on several markers, including urea, creatinine, pCS, and IS, in both their total and free forms.

## 4. Discussion

Preliminary evidence suggests that modulating the intestinal microbiota of CKD patients by oral supplementation with pre/probiotics or their combination (synbiotics) may decrease uremic toxins and inflammation, consequently improving endothelial function [29,30,33,34].

In renal replacement therapies, specifically in hemodialysis, Liu et al. [35] demonstrated that the consumption of probiotics did not significantly alter the species diversity of the fecal microbiome. However, it restored the community composition, with particular significance in non-diabetic HD patients (*p* = 0.007). Specifically, probiotics increased the proportions of the families Bacteroidaceae and Enterococcaceae. In contrast, they reduced the proportions of Ruminococcaceae, Halomonadaceae, Peptostreptococcaceae, Clostridiales Family XIII Incertae Sedis, and Erysipelotrichaceae in non-diabetic HD patients. Additionally, probiotics reduced the abundances of several uremic retention solutes in serum or feces, including indole-3-acetic acid-O-glucuronide, 3-guanidinopropionic acid, and 1-methylcytosine (*p* < 0.05). Also, in HD therapy, Cruz-Mora et al. [36] showed that synbiotic supplementation for 2 months significantly modified the intestinal microbiota by increasing Bifidobacteria counts in patients (*p* < 0.034).

On the other hand, in peritoneal dialysis therapy, Pan et al. [18] found that probiotics could significantly decrease the serum levels of high-sensitivity C-reactive protein and interleukin-6, which may mean that the consumption of probiotics favors both renal patients that are not undergoing replacement therapy and those that are.

However, there are studies in which synbiotic supplementation did not lower IS toxin levels in comparison with the placebo group (*p* = 0.438). Nevertheless, it had a major effect on improving constipation and the quality of life affected by constipation in patients undergoing chronic hemodialysis [37].

This is also the case with McFarlane et al. [34], who presented inconclusive results since no differences were observed between free and total uremic toxins between placebo and synbiotic groups in a population with stage 3–4 chronic kidney disease.

Interestingly, patients consuming probiotics trended to reduce the progression to end-stage renal disease and dialysis initiation, compared with subjects consuming placebo (cumulative survival 57% vs. 34%, log-rank *p* = 0.08 (Figure 2)). No differences between the two groups were, however, observed concerning the incidence of cardiovascular events (cumulative survival 81% vs. 89%, log-rank *p* = 0.55).

Current research also focused on gut microbiota alterations at various phases of CKD. It is important to explore the relationship between gut microbiota and CKD and investigate the difference in gut microbiota between patients with CKD and healthy individuals, as well as the different microbiota that could aggravate disease progression. Therefore, adjusting the use of probiotics, prebiotics, and synbiotics according to the distribution characteristics of floras at different phases has become a crucial measure to delay the course from CKD to ESRD.

All studies used various types of intervention delivery. The outcomes of the studies are controversial regarding the effects of microbiota-modulating agents on endothelial function. In addition, cardiovascular markers, inflammation, and the progression of chronic kidney disease should be noted.

### 4.1. Implications for Clinical Practice

The results from our study on modulating the gut microbiome in chronic kidney disease (CKD) patients have significant implications for clinical practice. The findings suggest that interventions using probiotics, prebiotics, and synbiotics can effectively reduce uremic toxins such as IS and pCS, which are known to exacerbate CKD and its complications.

Incorporating these microbiome-modulating interventions could become a valuable adjunct therapy for CKD management. By reducing uremic toxins, these interventions may help mitigate inflammation and oxidative stress, potentially slowing the progression of CKD and improving patient outcomes. This approach could be particularly beneficial for patients in stages 3 to 5 of CKD, where traditional therapies alone may not sufficiently address the accumulation of these toxins.

Furthermore, synbiotics, which combine probiotics and prebiotics, offer the most significant benefits, suggesting that a combined approach is more effective than using either component alone. This insight could guide healthcare providers in designing more effective dietary and therapeutic strategies for CKD patients.

### 4.2. Study Limitations

The study on the effects of probiotics, prebiotics, and synbiotics in CKD patients has several limitations that must be considered for a thorough understanding of the findings. Firstly, the sample size in the analyzed studies is limited, which increases the risk of type I errors and undermines the reliability of the results. Secondly, the heterogeneity of the studied population is limited; including patients at various stages of CKD and from the point of diagnosis could provide more comprehensive insights into the disease’s progression and the effectiveness of the interventions. Thirdly, while changes in the intestinal microbiota composition have demonstrated positive effects in reducing uremic toxins like pCS and IS, these changes are inconsistent across studies. This inconsistency underscores the need for more randomized clinical trials with larger sample sizes to confirm the therapeutic effects and establish reliable conclusions. Additionally, the study’s replicability and transparency are compromised by the lack of specific dates for the database searches, which hinders the ability to replicate the study’s methodology and independently verify the findings. Addressing these limitations in future research could significantly improve the understanding and management of CKD through gut microbiota modulation.

## 5. Conclusions

The results indicate promising potential, as studies have shown that altering the composition of patients’ intestinal microbiota can positively impact the reduction of pCS and IS. However, these findings are inconsistent, highlighting the need for more randomized clinical trials with larger sample sizes. More extensive studies are necessary to confirm the effects of these therapies on microbiota modulation and the reduction of uremic toxins, with broader detection than currently reported.

## Figures and Tables

**Figure 1 nutrients-17-01247-f001:**
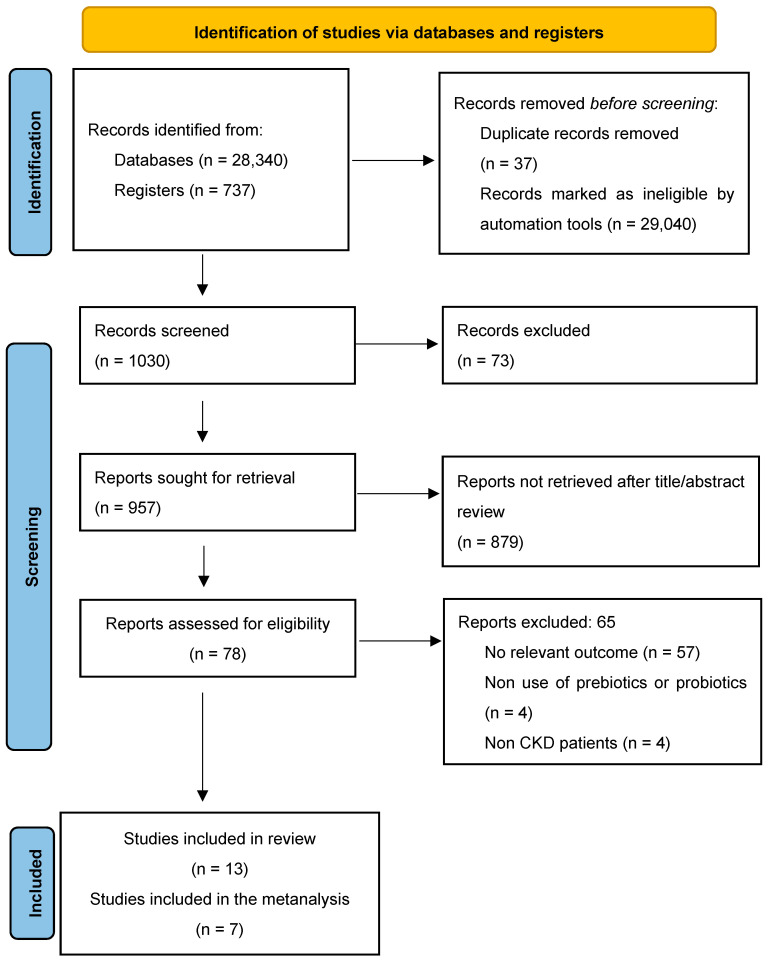
PRISMA 2020 flow diagram for new systematic reviews.

**Figure 2 nutrients-17-01247-f002:**
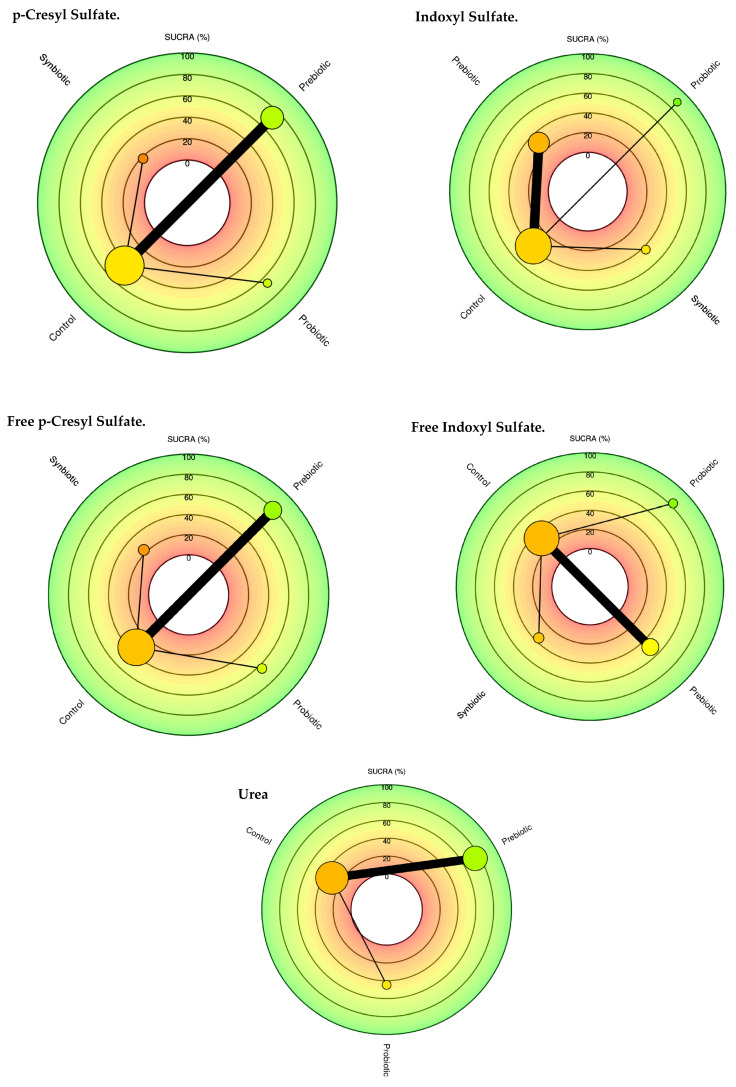
Network meta-analysis comparing probiotics, prebiotics, synbiotics, and control groups in reducing uremic toxins.

**Figure 3 nutrients-17-01247-f003:**
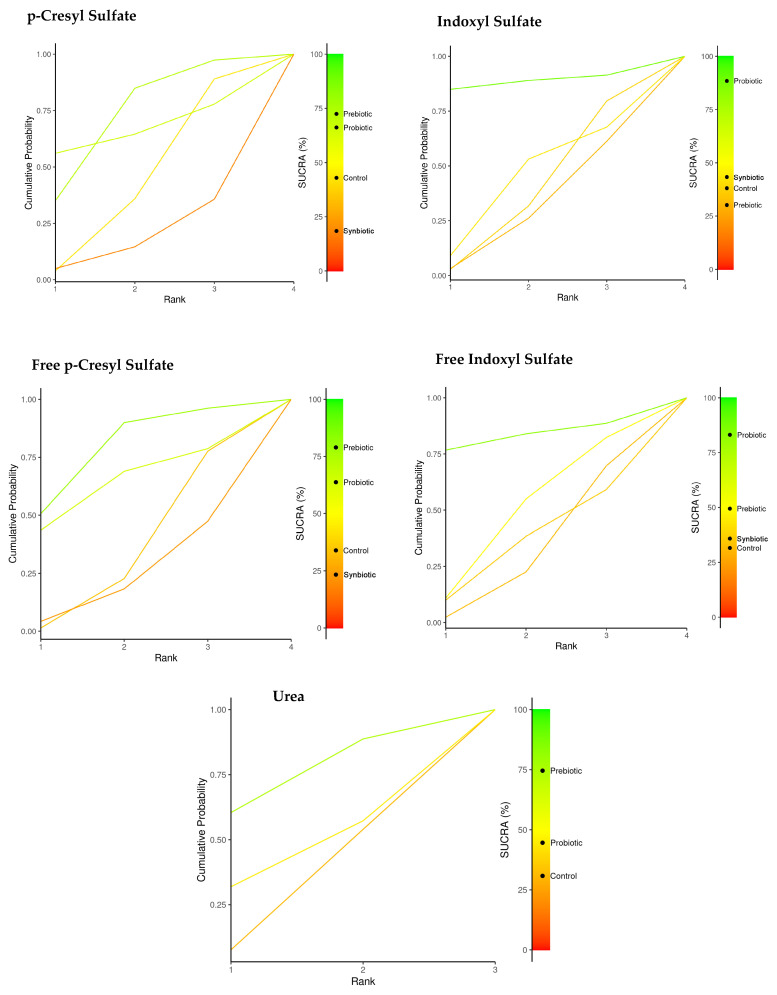
Network meta-analysis. Comparison of SUCRA values for probiotics, prebiotics, synbiotics, and the control group in reducing uremic toxins.

**Figure 4 nutrients-17-01247-f004:**
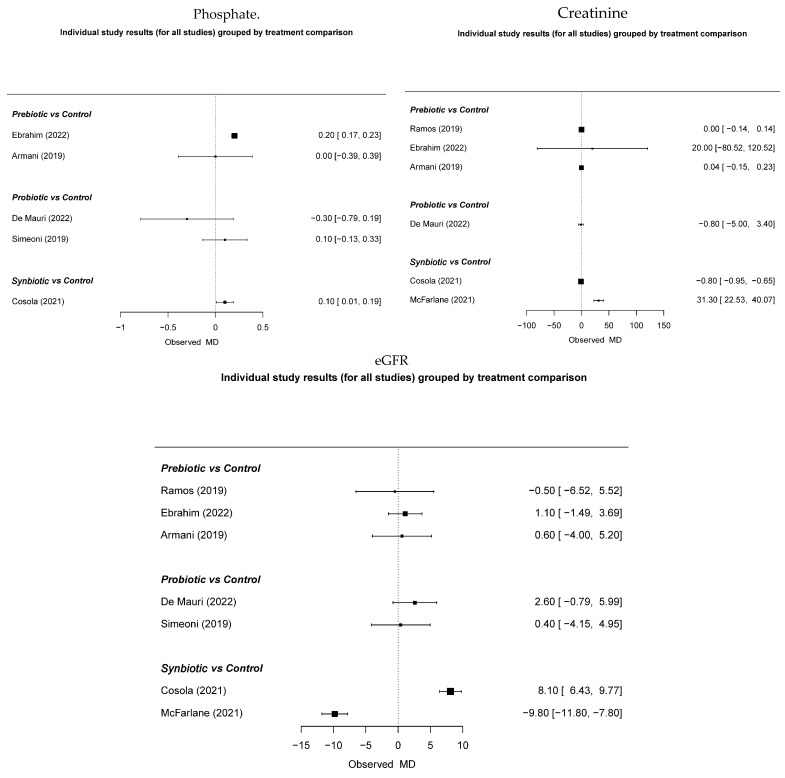
The effect of immunonutrition, creatinine, and glomerular filtration [19,28,29,31,32,33,34].

**Table 1 nutrients-17-01247-t001:** Characteristics of the studies included.

Author (Year) Country	Aim	Design	Sample Size	Intervention	Statistics Analysis	Main Results
De Mauri (2022) Italy [29]	Evaluate whether the association of selected probiotics on top of a low protein diet was able to reduce the burden of uremic, microbiota-derived, and proatherogenic toxins in patients with advanced renal failure who were not on dialysis.	Placebo-controlled, randomized study	IG = 24 CG = 23	Probiotic	Analysis of blood and urine samples. Shapiro–Wilk and q–q plot tests, Mann–Whitney U-test, Wilcoxon signed rank test and Kaplan–Meier analysis	Patients in the placebo group showed increased serum values of total cholesterol, LDL cholesterol, lipoprotein-associated phospholipase, and IS, while the 24 subjects in the probiotics group showed a trend in the reduction of microbiota toxins. A reduction of antihypertensive and diuretic medications was possible in the probiotics group.
Cosola (2021) Italy [33]	Investigate the effects of the synbiotic on the barrier permeability of different gastrointestinal tracts and on gastrointestinal symptoms assessed by the GSRS questionnaire.	Randomized trial	IG = 23 CG = 24	Synbiotic	Urine and blood samples Linear regression analysis. Mann–Whitney test, Kruskal–Wallis multiple-comparison, Spearman, Fisher’s LSD.	Two-month administration of the synbiotic resulted in a decrease in free IS in the CKD group. After supplementation, reduction of small intestinal permeability and amelioration of abdominal pain and constipation syndromes were observed only in the CKD group.
McFarlane (2021) Australia [34]	Evaluate the feasibility of a trial of long-term synbiotic supplementation in adults with stage 3–4 CKD.	Double-blind, placebo-controlled, randomized trial	IG = 28 CG = 28	Synbiotic	Blood and stool samples were collected. Chi-square, Fisher’s exact tests, Student’s t-test, and Mann–Whitney test	No differences were observed between free and total uremic toxins between placebo and synbiotic groups. Synbiotic supplementation resulted in a 3.14 mL/min/1.73 m^2^ reduction in eGFR and a 20.8 µmol/L increase in serum creatinine concentration.
Ramos (2019) Brazil [31]	Investigate the effect of a prebiotic fructooligosaccharide (FOS) on uremic toxins of non-dialysis-dependent CKD (NDD-CKD) patients.	Randomized trial	IG = 23 CG = 23	Prebiotic	Blood samples were collected. Shapiro–Wilk test, chi-square, Fisher’s exact tests, Student’s t-test, and Mann–Whitney test.	There was a trend in the difference of serum total DPCS and serum-free Δ%PCS between the groups. Aside from the decreased high-density lipoprotein cholesterol in the intervention, no differences were observed in the change in IS, IAA, or other secondary outcome between the groups.
Simeoni (2019) Italy [28]	Describe gut dysbiosis in initial stages of CKD.	Clinical trial	IG = 24 CG = 23	Probiotic	Blood, urine, and stool samples were collected. Student’s paired t test, Wilcoxon test, simple t test, Mann–Whitney U test, Pearson.	Mean fecal Lactobacillales and Bifidobacteria concentrations were increased only in the probiotics group. Conversely, mean urinary indican and 3-MI levels increased only in the group treated with probiotics. Compared to placebo group, significant improvements in C-reactive protein, iron, ferritin, transferrin saturation, β2-microglobulin, serum iPTH, and serum calcium were observed only in the probiotics group.
Ebrahim (2022) South Africa [32]	Investigate the effect of a ß-glucan prebiotic on kidney function, uremic toxins, and the gut microbiome in stage 3 to 5 CKD participants.	Randomized control trial	IG = 23 CG = 22	Prebiotic	Blood samples for quantification of uremic toxins and stool samples for characterization of the gut microbiome were obtained. Kolmogorov, Mann–Whitney and chi-square tests.	There was a significant reduction in uremic toxin levels at different time points, in free IS at 8 weeks and 14 weeks, free pCS at 14 weeks, and total and free pCG. There were no differences in relative abundances of genera between groups. The ß-glucan prebiotic significantly altered uremic toxin levels of intestinal origin and favorably affected the gut microbiome.
Armani (2022) Brazil [19]	Evaluate the effect of the prebiotic FOS on endothelial function and arterial stiffness in non-dialysis CKD patients.	Randomized controlled trial	IG = 23 CG = 23	Prebiotic	Fasting blood samples were collected. Kolmogorov–Smirnov, Student’s t-test, Mann–Whitney U-test, Wilcoxon, chi-squared analysis, Fischer’s test, and McNemar’s test.	There was a significant decrease in IL-6 levels and a trend toward pCS reduction only in the prebiotic group. Comparing both groups, there was no difference in FMD and PWV. FMD remained stable in the prebiotic group, while it decreased in the placebo group.

CKD: chronic kidney disease, pCS: p-Cresyl Sulfate; IS: Indoxyl Sulfate; IAA: indole-3-acetic acid; LPS: lipopolysaccharide; FOS: fructooligosaccharide; IL-6: interleukin-6; iPTH: intact parathyroid hormone; FMD: flow-mediated dilatation; PWV: pulse wave velocity.

**Table 2 nutrients-17-01247-t002:** SUCRA values for each uremic toxin are combined by category.

	Probiotics	Prebiotics	Synbiotics	Control
P-Cresyl Sulfate	66.2	**72.4**	18.4	43.0
Free p-Cresyl	63.8	**78.9**	23.3	34.0
Total Indoxyl Sulfate	**88.5**	30.2	43.3	38.1
Free Indoxyl Sulfate	**83.1**	49.4	35.9	31.6
Urea	44.6	**74.6**	--	30.8
Creatinine	60.5	56.9	19.4	63.2
Phosphate	67.8	21.1	39.3	71.8
eGFR	47.9	39.0	59.7	53.4
All-Outcomes-Combined	**65.3**	**52.8**	39.5	45.7

Bold values indicate the highest SUCRA value for each uremic toxin category, representing the most effective intervention among the compared groups.

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
