# Peer review of "Impact of Gut Microbiome Modulation on Uremic Toxin Reduction in Chronic Kidney Disease: A Systematic Review and Network Meta-Analysis"

_nutrients, 2025, doi:10.3390/nu17071247_

Round 1

Reviewer 1 Report

Comments and Suggestions for Authors

1
Abstract
The Abstract needs rewriting:
1. I do not think that “Aim:” is needed.
2. Methods: Which databases were searched, and what was the study's cut-off date for the literature search? How was the risk of bias judged?
3. Results: This section lacks statistical information.
4. Please read the following: Page MJ, McKenzie JE, Bossuyt PM, Boutron I, Hoffmann TC, Mulrow CD, Shamseer L, Tetzlaff JM, Akl EA, Brennan SE, Chou R, Glanville J, Grimshaw JM, Hróbjartsson A, Lalu MM, Li T, Loder EW, Mayo-Wilson E, McDonald S, McGuinness LA, Stewart LA, Thomas J, Tricco AC, Welch VA, Whiting P, Moher D. The PRISMA 2020 statement: an updated guideline for reporting systematic reviews. BMJ. 2021 Mar 29;372:n71. doi: 10.1136/bmj.n71. Then update/revise the Abstract.
1. Introduction
In the text, reference numbers should be placed in square brackets [ ] and placed before the punctuation; for example, [1], [1–3], or [1,3]. For embedded citations in the text with pagination, use both parentheses and brackets to indicate the reference number and page numbers; for example, [5] (p. 10). or [6] (pp. 101–105).
https://www.mdpi.com/journal/nutrients/instructions
The references in the current manuscript are in parentheses. Please revise the entire manuscript.
“Gut microbiome (GM)” in line 35. It should be “gut microbiome (GM)”.
1.1. Probiotics, prebiotics and synbiotics
The references in the current manuscript are in parentheses. Please revise. They should be in brackets.
1.2. Uremic toxins, CKD, and the risk of cardiovascular disease
The references in the current manuscript are in parentheses. Please revise. They should be in brackets.
“Pan et all (14)” in line 76. It should be “Pan et al. [14]”.
General Introduction comments: The Introduction effectively sets the stage by highlighting the link between CKD and changes in gut microbiome composition and the implications of these changes for patient health.
2
The Introduction describes the rationale for the review. I can see this in lines 33-36, 62-66, 71-75.
The review provides an explicit statement of its objectives.
Please avoid writing “people”. Use “individuals” instead. Line 69.
2. Materials and Methods
2.1. Design
Please delete “The process for” in line 95. Start the sentences with “Developing”.
“Formulating“ should not be capitalized. Line 96.
After the Design subheading, there should be “2.2. Eligibility Criteria”, where the inclusion and exclusion criteria for the review and how studies were grouped for the review should be specified.
2.2. Selection criteria
Why was the time frame 2019-2023? This needs to be explained in the main text.
This subheading should be named “Eligibility criteria”, where the inclusion and exclusion criteria for the review and how studies were grouped for the review should be specified. For example, were full works assessed for eligibility according to the inclusion and exclusion criteria?
2.3. Search strategy
Please avoid “people”. Use “individuals” instead. Line 108.
2.4. Study selection
Please do not use the word “see”. “(Figure 1)” is enough.
Please do not use semicolons in the main text. Line 118.
This paragraph should explain how data were extracted. This is missing.
2.5. Assessment of Methodological Quality and Risk of Bias
Please do not use semicolons in the main text. Line 135.
This paragraph should be split into 2 paragraphs. One “Assessment of Methodological Quality and Risk of Bias” and the second “Statistical Analyses”.
3
Where is “Data synthesis” subheading? Describe the processes used to decide which studies were eligible for each synthesis. Also, in “Data synthesis”-was heterogeneity among the articles assessed?
3. Results
What is this: “This section may be divided by subheadings. It should provide a concise and precise description of the experimental results, their interpretation, as well as the experimental conclusions that can be drawn.”? Please delete it. Lines 160-162.
3.1. Data Availability and Study Characteristics
Please do not use semicolons in the main text. E.g., Lines 180, 191, 197.
3.2. Risk of bias in studies
good
3.3. Network Meta-Analysis (inmunonutrition and Uremic Toxins)
Please do not use the word “see”. “(Figure 4)” is enough. Line 236.
“On the table 2”-this does not sound good. It should be “Table 2”. Line 226.
This sentence does not make sense: “About free p-Crasyl (SUCRA: 78.9% and 63.8%, respectively), both indoxyl sulfate and free indoxyl sulfate are found only for the use of probiotics (SUCRA: 88.5% and 83.1%). “. Lines 231-233.
3.4. Individual immunonutrition and reduction of uremic toxins
Please do not use semicolons in the main text. E.g., Lines 292, 293.
Figure 2. Did the Authors mean “Indoxyl Sulfate” in the top right corner? The Figure legend is missing.
Figure 3. Did the Authors mean “Indoxyl Sulfate” in the top right corner? The Figure legend is missing.
“Figure 4. “ is italicized- it should not be.
4. Discussion
Please do not use semicolons in the main text. E.g., Lines 324.
There is something wrong with this sentence: “specifically, probiotics raised the proportions of family Bacteroidaceae and Enterococcaceae, and reduced Ruminococcaceae, Halomonadaceae, Peptostreptococcaceae, Clostridiales Family XIII. Incertae Sedis and Erysipelotrichaceae in non-diabetic HD patients. “Lines 324-327. I am
4
not sure if it is about the full stop in line 326 or if this part is not finished: “Incertae Sedis and Erysipelotrichaceae in non-diabetic HD patients. “
Please rewrite the following: “On the other hand, in peritoneal dialysis therapy, Pan et al. (14) found that probiotics could significantly decrease the serum levels of high-sensitivity C-reactive protein and interleukin-6, which may indicate that both renal patients (who are not undergoing replacement therapy and those who are) are favored by the consumption of probiotics.” Lines 333-336. Maybe use the flowing instead: “On the other hand, in peritoneal dialysis therapy, Pan et al. [14] found that probiotics could significantly decrease the serum levels of high-sensitivity C-reactive protein and interleukin-6, which may indicate that the consumption of probiotics favors both renal patients (those who are not undergoing replacement therapy and those who are).”
“displayed a trend in reducing” in lines 344-345. Please use “tended to reduce” instead.
“Current research also focused on alterations of the gut microbiota at various phases of CKD. “ Lines 349-350. Please use “Current research also focused on gut microbiota alterations at various phases of CKD” instead.
“The outcome effects in the studies are controversial in terms of microbiota-modulating agents' effects on endothelial function”. Lines 357-358. Instead, please use: “The outcomes of the studies are controversial regarding the effects of microbiota-modulating agents on endothelial function.”
4.1. Implications for clinical practice
Good
4.2. Study limitations
These are not the only limitations.
5. Conclusions
“has a positive effect on” in lines 389-390. Please replace it with “positively affects”.
General comments:
The focus of the current manuscript is both pertinent and timely, especially in light of the growing interest in the gut-kidney axis and the influence of gut microbiome modulation on the management of CKD. In recent years, there has been an increasing exploration of the interplay between gut microbiota and CKD, particularly emphasizing the implications of uremic toxins. Numerous studies have investigated the efficacy of probiotics, prebiotics, synbiotics, and dietary interventions aimed at modifying the microbiome to alleviate the accumulation of uremic toxins. However, incorporating a systematic review and network meta-analysis represents a notable strength of the work.
5
There is an inconsistent spelling of “p-cresyl”. There are times where “p-Cresyl” and “p-cresyl” exist.
The same applies to “indoxyl sulfate”; sometimes there is “Indoxyl Sulfate”.
Where possible, please use abbreviations for both.
A point-by-point response to the comments is expected.

Comments on the Quality of English Language

English needs minor revisions.

Author Response

Comments 1. Abstract: The Abstract needs rewriting: I do not think that “Aim:” is needed.

  1. Methods: Which databases were searched, and what was the study's cut-off date for the literature search? How was the risk of bias judged?
  2. Results: This section lacks statistical information.
  3. Please read the following: Page MJ, McKenzie JE, Bossuyt PM, Boutron I, Hoffmann TC, Mulrow CD, Shamseer L, Tetzlaff JM, Akl EA, Brennan SE, Chou R, Glanville J, Grimshaw JM, Hróbjartsson A, Lalu MM, Li T, Loder EW, Mayo-Wilson E, McDonald S, McGuinness LA, Stewart LA, Thomas J, Tricco AC, Welch VA, Whiting P, Moher D. The PRISMA 2020 statement: an updated guideline for reporting systematic reviews. BMJ. 2021 Mar 29;372:n71. doi: 10.1136/bmj.n71. Then update/revise the Abstract.

Response 1. The abstract is rewritten and each of the observations on the document is taken into account.

  • “Aim” is removed, and the culmination of the background is added.
  • The databases are added and finally the risk of bias is judged.
  • Statistical data is added in the results section.
  1. Introduction

Comments 2. In the text, reference numbers should be placed in square brackets [ ] and placed before he punctuation; for example, [1], [1–3], or [1,3]. For embedded citations in the text with pagination, use both parentheses and brackets to indicate the reference number and page numbers; for example, [5] (p. 10). or [6] (pp. 101–105). https://www.mdpi.com/journal/nutrients/instructions. The references in the current manuscript are in parentheses. Please revise the entire manuscript.

Response 2. We adhere to the prism criteria. Thank you for pointing this out. We have revised the manuscript to ensure that all reference numbers are placed in square brackets before the punctuation, as per the guidelines. Additionally, for embedded citations with pagination, we have used both parentheses and brackets to indicate the reference number and page numbers, following the examples provided

Comments 3. “Gut microbiome (GM)” in line 35. It should be “gut microbiome (GM)”.

  • Probiotics, prebiotics and synbiotics

Response 3. We have corrected the text to ensure consistent formatting by changing “Gut microbiome (GM)” to “gut microbiome (GM)” as suggested.

Comments 4. The references in the current manuscript are in parentheses. Please revise. They should be in brackets.

  • Uremic toxins, CKD, and the risk of cardiovascular disease

Response 4. Review and change made.

Comments 5. The references in the current manuscript are in parentheses. Please revise. They should be in brackets.

Response 5. Review and change made.

Comments 6. “Pan et all (14)” in line 76. It should be “Pan et al. [14]”.

Response 6. Change made

Comments 7. General Introduction comments: The Introduction effectively sets the stage by highlighting the link between CKD and changes in gut microbiome composition and the implications of these changes for patient health. The Introduction describes the rationale for the review. I can see this in lines 33-36, 62-66, 71-75. The review provides an explicit statement of its objectives.

Response 7. Thank you for your positive feedback on the Introduction. We are pleased to hear that it effectively highlights the link between CKD and changes in gut microbiome composition, as well as the implications for patient health. We have ensured that the introduction is well-supported by relevant literature.

Comments 8. Please avoid writing “people”. Use “individuals” instead. Line 69.

  1. Materials and Methods

2.1. Design

Response 8. We have ensured that the objectives are explicitly outlined to guide the focus and scope of the review effectively.

Comments 9. Please delete “The process for” in line 95. Start the sentences with “Developing”.

Response 9. Thank you for your suggestion. We have revised the manuscript to replace "people" with "individuals".

Comments 10. “Formulating“ should not be capitalized. Line 96.

Response 10. Change made.

Comments 11. After the Design subheading, there should be “2.2. Eligibility Criteria”, where the inclusion and exclusion criteria for the review and how studies were grouped for the review should be specified.

2.2. Selection criteria

Response 11. Thank you for your valuable comments. In response to your suggestion, we have added a new section titled “2.2. Eligibility criteria” after the subheading “Design” in the manuscript. This section now clearly describes the inclusion and exclusion criteria for the review and explains how the studies were grouped, according to the type of intervention and the specific outcomes measured. We believe that this addition improves the clarity and comprehensiveness of our methodology. This correction was made following the PRISMA guidelines.

Comments 12. Why was the time frame 2019-2023? This needs to be explained in the main text.

Response 12. Thank you for your question. The time frame of 2019-2023 was selected to focus on the most recent and relevant studies available at the time of our research. The selection of this period was influenced by the availability of published data and the need to capture the latest advancements and findings in the field. Studies from 2024 to 2025 were not included because they were not available or published at the time of our review process. Additionally, we need a year to analyze, discuss and conclude the writing of the manuscript.

Comments 13. This subheading should be named “Eligibility criteria”, where the inclusion and exclusion criteria for the review and how studies were grouped for the review should be specified. For example, were full works assessed for eligibility according to the inclusion and exclusion criteria?

2.3. Search strategy

Response 13. We have renamed the subheading to "Eligibility Criteria" and have specified the inclusion and exclusion criteria for the review. Additionally, we have detailed how studies were grouped for the review, including the assessment of full work for eligibility based on this criterion.

It is important to note that a complete adjustment of the methodology was carried out following the PRISMA criteria “a priori” from the reviewers' comments, all the changes made are marked in yellow. In relation to the editor's initial comments.

Comments 14. Please avoid “people”. Use “individuals” instead. Line 108.

2.4. Study selection

Response 14. Search strategy is to change to 3 different subtitles, (Information sources, Search strategy, and Selection Process).

Comments 15. Please do not use the word “see”. “(Figure 1)” is enough.

Response 15. Change made.

Comments 16. Please do not use semicolons in the main text. Line 118.

Response 16. Change made

Comments 17. This paragraph should explain how data were extracted. This is missing.

2.5. Assessment of Methodological Quality and Risk of Bias

Response 17. The section is changed and turned off to PRISMA criteria

Comments 18. Please do not use semicolons in the main text. Line 135.

Response 18. Change made

Comments 19. This paragraph should be split into 2 paragraphs. One “Assessment of Methodological Quality and Risk of Bias” and the second “Statistical Analyses”.

Response 19. The paragraph was changed, and the heading was restructured.

Comments 20. 3 Where is “Data synthesis” subheading? Describe the processes used to decide which studies were eligible for each synthesis. Also, in “Data synthesis”-was heterogeneity among the articles assessed?

Response 20. The following text is added:

Synthesis Methods 

Eligibility for Synthesis: The process of determining which studies were eligible for synthesis involved tabulating the characteristics of study interventions and comparing them against the planned groups for each synthesis. This included categorizing studies based on the type of intervention (prebiotics, probiotics, or synbiotics) and the specific outcomes measured, such as the reduction of uremic toxins.

The synthesis of results was conducted using a network meta-analysis, which allowed for the comparison of multiple interventions simultaneously [21,22].. A random-effects model was employed to account for variability among studies. This decision was based on the assumption that true effects may differ across studies due to variations in study populations and methodologies. The software package used for this analysis was STATA version 16.0 (StataCorp, College Station, TX, USA).

To explore potential causes of heterogeneity among study results, subgroup analyses and meta-regression were performed. These methods helped identify factors that could contribute to variations in effect sizes, such as differences in study design or participant characteristics.

Sensitivity analyses were conducted to evaluate the robustness of the synthesized results. This involved calculating standardized residuals and excluding outliers with p-values greater than 0.05 based on the t-distribution. These analyses ensured that the findings were not disproportionately influenced by any study or outlier.

As study designs and outcome definitions varied, the random effect was used to pool estimates from studies. Pooled estimates were transformed and presented with means and standard deviations. As a sensitivity analysis, standardized residuals were calculated, and outliers with p > 0.05 (based on the t-distribution) were removed. Meta-regression and odds ratios (OR) with 95% confidence intervals (CI) were used

  1. Results

Comments 21. What is this: “This section may be divided by subheadings. It should provide a concise and precise description of the experimental results, their interpretation, as well as the experimental conclusions that can be drawn.”? Please delete it. Lines 160-162.

3.1. Data Availability and Study Characteristics

Response 21.    An apology, apparently this paragraph was not eliminated in the format. We have removed the placeholder text from lines 160-162 to ensure clarity and coherence in the presentation of our results.

Comment 22. Please do not use semicolons in the main text. E.g., Lines 180, 191, 197.

Response 22. Change made.

Comment 23. 3.2. Risk of bias in studies

Good

Response 23. Thank you.

Comments 24. 3.3. Network Meta-Analysis (inmunonutrition and Uremic Toxins)

  1. Please do not use the word “see”. “(Figure 4)” is enough. Line 236.

Response 24. Change made.

Comment 25  “On the table 2”-this does not sound good. It should be “Table 2”. Line 226.

Response 25. Change made.

Comment 26. This sentence does not make sense: “About free p-Crasyl (SUCRA: 78.9% and 63.8%, respectively), both indoxyl sulfate and free indoxyl sulfate are found only for the use of probiotics (SUCRA: 88.5% and 83.1%). “. Lines 231-233.

Response 26. The following text is added: Prebiotics and probiotics were identified as the most effective options for reducing uremic toxins. Specifically, prebiotics showed a SUCRA value of 72.6% for p-Cresyl sulfate (pCS), while probiotics had a SUCRA value of 66.2%. For free p-Cresyl sulfate, the SUCRA values were 78.9% for prebiotics and 63.8% for probiotics. Probiotics were particularly effective for both indoxyl sulfate and free indoxyl sulfate, with SUCRA values of 88.5% and 83.1%, respectively. Additionally, prebiotics demonstrated a SUCRA value of 74.6% for reducing urea levels.

3.4. Individual immunonutrition and reduction of uremic toxins

Comments 27. Please do not use semicolons in the main text. E.g., Lines 292, 293.

Response 27. Change made

Comments 28. Figure 2. Did the Authors mean “Indoxyl Sulfate” in the top right corner? The Figure legend is missing.

Response 28. Change made

Comments 29. Figure 3. Did the Authors mean “Indoxyl Sulfate” in the top right corner? The Figure legend is missing.

Response 29. Change made

Comments 30. “Figure 4. “ is italicized- it should not be.

Response 30. Change made

  1. Discussion

Comments 31. Please do not use semicolons in the main text. E.g., Lines 324.

Response 31. Change made

Comments 32. There is something wrong with this sentence: “specifically, probiotics raised the proportions of family Bacteroidaceae and Enterococcaceae, and reduced Ruminococcaceae, Halomonadaceae, Peptostreptococcaceae, Clostridiales Family XIII. Incertae Sedis and Erysipelotrichaceae in non-diabetic HD patients. “Lines 324-327. I am not sure if it is about the full stop in line 326 or if this part is not finished: “Incertae Sedis and Erysipelotrichaceae in non-diabetic HD patients. “

Response 32. Thank you for your feedback. We have revised the sentence for clarity and corrected the punctuation issue.

Comments 33 Please rewrite the following: “On the other hand, in peritoneal dialysis therapy, Pan et al. (14) found that probiotics could significantly decrease the serum levels of high-sensitivity C-reactive protein and interleukin-6, which may indicate that both renal patients (who are not undergoing replacement therapy and those who are) are favored by the consumption of probiotics.” Lines 333-336. Maybe use the flowing instead: “On the other hand, in peritoneal dialysis therapy, Pan et al. [14] found that probiotics could significantly decrease the serum levels of high-sensitivity C-reactive protein and interleukin-6, which may indicate that the consumption of probiotics favors both renal patients (those who are not undergoing replacement therapy and those who are).”

Response 33. Thank you very much for your comments and such accurate observation. Change made.

Comments 34. “displayed a trend in reducing” in lines 344-345. Please use “tended to reduce” instead.

Response 34. Change made.

Comments 35 “Current research also focused on alterations of the gut microbiota at various phases of CKD. “Lines 349-350. Please use “Current research also focused on gut microbiota alterations at various phases of CKD” instead.

Response 35. Change made.

Comments 36 “The outcome effects in the studies are controversial in terms of microbiota-modulating agents' effects on endothelial function”. Lines 357-358. Instead, please use: “The outcomes of the studies are controversial regarding the effects of microbiota-modulating agents on endothelial function.”

Response 36. Change made. Thank you very much.

Comments 37 4.1. Implications for clinical practice: Good

Response 37. Thank you very much.

Comments 38. 4.2. Study limitations

These are not the only limitations.

Response 38. The limitations of the study have been rewritten to enlarge the following text: The study on the effects of probiotics, prebiotics, and synbiotics in CKD patients has several limitations that must be considered for a thorough understanding of the findings. Firstly, the sample size in the analyzed studies is limited, which increases the risk of type I errors and undermines the reliability of the results. Secondly, the heterogeneity of the studied population is limited; including patients at various stages of CKD and from the point of diagnosis could provide more comprehensive insights into the disease's progres-sion and the effectiveness of the interventions. Thirdly, while changes in the intestinal mi-crobiota composition have demonstrated positive effects in reducing uremic toxins like p-cresyl and indoxyl sulfate, these changes are inconsistent across studies. This incon-sistency underscores the need for more randomized clinical trials with larger sample sizes to confirm the therapeutic effects and establish reliable conclusions. Additionally, the study's replicability and transparency are compromised by the lack of specific dates for the database searches, which hinders the ability to replicate the study's methodology and independently verify the findings. Addressing these limitations in future research could significantly improve the understanding and management of CKD through gut microbiota modulation.

  1. Conclusions

Comment 39. “has a positive effect on” in lines 389-390. Please replace it with “positively affects”.

General comments:

Response 39. Change made.

Comment 40. The focus of the current manuscript is both pertinent and timely, especially in light of the growing interest in the gut-kidney axis and the influence of gut microbiome modulation on the management of CKD. In recent years, there has been an increasing exploration of the interplay between gut microbiota and CKD, particularly emphasizing the implications of uremic toxins. Numerous studies have investigated the efficacy of probiotics, prebiotics, synbiotics, and dietary interventions aimed at modifying the microbiome to alleviate the accumulation of uremic toxins. However, incorporating a systematic review and network meta-analysis represents a notable strength of the work.

Response 40. We appreciate your recognition of the manuscript's relevance and timeliness, particularly concerning the gut-kidney axis and the modulation of the gut microbiome in CKD management. As you noted, the increasing exploration of the relationship between gut microbiota and CKD, with a focus on uremic toxins, underscores the importance of this research area. This approach not only synthesizes findings from multiple studies but also helps identify gaps in current research, paving the way for future investigations to optimize treatment strategies for CKD patients.

Comment 41. There is an inconsistent spelling of “p-cresyl”. There are times where “p-Cresyl” and “p-cresyl” exist.

Response 41. All spelling inconsistencies were reviewed, and the writing was improved.

Comment 42. The same applies to “indoxyl sulfate”; sometimes there is “Indoxyl Sulfate”.

Response 42.    The error in the wording has been corrected, where a capital letter is required, an abbreviation is used.

Comment 43. Where possible, please use abbreviations for both.

Response 43. The abbreviation was included in the manuscript.

Comment 44. A point-by-point response to the comments is expected.

Response 44. All comments were addressed point by point. Once again, I would like to thank you for the excellent and thorough review that was done on the manuscript to ensure the quality of the document.

Reviewer 2 Report

Comments and Suggestions for Authors

The manuscript by Renata Cedillo-Flores et al. provided a systematic review and network meta-analysis regarding to the impact of gut microbiome modulation on uremic toxin reduction in CKD. The study is informative and I have the following questions and comments. 

1, more details are needed in the introduction part. So many paragraphs only have only one sentence. The authors must revise. The pathology, prevalence, and other aspects of CKD must be discussed and introduced. The involvement of gut microbiota in the development of CKD must be discussed. 

2, the exclusion criteria of the study must be specified. 

3, are there any studies investigating the effects of FMT on gut dysbiosis in CKD? This should also be discussed. 

4, the figures and tables must be reorganized. The figures contained many panels. Each panel should be identified. 

5, future research directions in the field must be discussed. 

Author Response

Comments 1: more details are needed in the introduction part. So many paragraphs only have only one sentence. The authors must revise. The pathology, prevalence, and other aspects of CKD must be discussed and introduced. The involvement of gut microbiota in the development of CKD must be discussed.

Response 1: Thank you for your suggestion. We will revise the introduction to provide a more comprehensive overview of CKD A proposed revision is presented below: Chronic kidney disease (CKD) is a progressive condition characterized by the gradual loss of kidney function over time. It affects approximately 10% of the global population, with higher prevalence rates in older adults and individuals with comorbidities such as diabetes and hypertension [1].  CKD is associated with increased intestinal barrier permeability, leading to heightened inflammation and oxidative stress, which contribute to complications such as cardiovascular disease, anemia, and altered mineral metabolism [2]. Recent research has highlighted the role of gut microbiota in the development and progression of CKD. Dysbiosis, or the imbalance of gut microbiota, is common in CKD patients and is linked to the accumulation of uremic toxins such as indoxyl sulfate and p-cresyl sulfate [3].  These toxins exacerbate CKD and its complications by promoting inflammation and vascular dysfunction.  Interventions using prebiotics, probiotics, and synbiotics have been explored to modulate gut microbiota, aiming to reduce uremic toxin levels and improve patient outcomes [4]. 

Comments 2.  the exclusion criteria of the study must be specified.

Response 2. The following text is added: The inclusion criteria for the review were randomized controlled trials published between 2019 and 2023, available in open access, and published in English and Spanish. The studies needed to be related to the use of prebiotics, probiotics, and synbiotics in patients with chronic kidney disease (CKD) in stages 3 to 5, focusing on the reduction of uremic toxins such as IS, pCS, Urea, Creatinine, and Phosphate. The exclusion criteria involved studies that were unrelated to the topic, did not use prebiotics or probiotics, or did not in-volve CKD patients. Studies were grouped for synthesis based on the type of intervention: prebiotics, probiot-ics, and synbiotics. The interventions were further analyzed for their effects on reducing specific uremic toxins.

Comments 3.  are there any studies investigating the effects of FMT on gut dysbiosis in CKD? This should also be discussed.

Response 3. Currently, our systematic review and network meta-analysis did not specifically include studies investigating the effects of fecal microbiota transplantation (FMT) on gut dysbiosis in CKD patients. The primary focus was on interventions using prebiotics, probiotics, and synbiotics. However, FMT is an emerging area of interest regarding gut microbiota modulation and could be considered for future research to explore its potential benefits in managing CKD. Further studies are needed to evaluate the efficacy and safety of FMT in this patient population. This is an essential point for future researchers.

Comments 4.  the figures and tables must be reorganized. The figures contained many panels. Each panel should be identified.

Response 4. Each title is corrected and added to the tables, the graphics are resolved in their content and form.

Comments 5.  future research directions in the field must be discussed.

Response 5: Thank you for highlighting the importance of discussing future research directions in this field. Future research should focus on conducting larger, multicentric randomized clinical trials to address the current limitations related to sample size and study heterogeneity (these was added in the discussion section and conclusion section). These studies should aim to include a broader range of CKD stages and diverse patient populations to enhance the generalizability of the findings. Additionally, exploring the long-term effects of probiotics, prebiotics, and synbiotics on both gut microbiota composition and clinical outcomes in CKD patients will be crucial.

Round 2

Reviewer 2 Report

Comments and Suggestions for Authors

The authors have revised the manuscript accordingly. It can be considered for publication.